# Geometric Matrix Completion with Recurrent Multi-Graph Neural Networks

**Federico Monti**
Università della Svizzera italiana
Lugano, Switzerland
federico.monti@usi.ch

**Michael M. Bronstein**
Università della Svizzera italiana
Lugano, Switzerland
michael.bronstein@usi.ch

**Xavier Bresson**
School of Computer Science and Engineering
NTU, Singapore
xbresson@ntu.edu.sg

## Abstract

Matrix completion models are among the most common formulations of recommender systems. Recent works have showed a boost of performance of these techniques when introducing the pairwise relationships between users/items in the form of graphs, and imposing smoothness priors on these graphs. However, such techniques do not fully exploit the local stationary structures on user/item graphs, and the number of parameters to learn is linear w.r.t. the number of users and items. We propose a novel approach to overcome these limitations by using geometric deep learning on graphs. Our matrix completion architecture combines a novel multi-graph convolutional neural network that can learn meaningful statistical graph-structured patterns from users and items, and a recurrent neural network that applies a learnable diffusion on the score matrix. Our neural network system is computationally attractive as it requires a constant number of parameters independent of the matrix size. We apply our method on several standard datasets, showing that it outperforms state-of-the-art matrix completion techniques.

## 1 Introduction

Recommender systems have become a central part of modern intelligent systems. Recommending movies on Netflix, friends on Facebook, furniture on Amazon, and jobs on LinkedIn are a few examples of the main purpose of these systems. Two major approaches to recommender systems are collaborative [5] and content [32] filtering techniques. Systems based on collaborative filtering use collected ratings of items by users and offer new recommendations by finding similar rating patterns. Systems based on content filtering make use of similarities between items and users to recommend new items. Hybrid systems combine collaborative and content techniques.

**Matrix completion.** Mathematically, a recommendation method can be posed as a *matrix completion* problem [9], where columns and rows represent users and items, respectively, and matrix values represent scores determining whether a user would like an item or not. Given a small subset of known elements of the matrix, the goal is to fill in the rest. A famous example is the Netflix challenge [22] offered in 2009 and carrying a 1M\$ prize for the algorithm that can best predict user ratings for movies based on previous user ratings. The size of the Netflix matrix is 480k movies × 18k users (8.5B entries), with only 0.011% known entries.

Recently, there have been several attempts to incorporate geometric structure into matrix completion problems [27, 19, 33, 24], e.g. in the form of column and row graphs representing similarity of users

and items, respectively. Such additional information defines e.g. the notion of *smoothness* of the matrix and was shown beneficial for the performance of recommender systems. These approaches can be generally related to the field of *signal processing on graphs* [37], extending classical harmonic analysis methods to non-Euclidean domains (graphs).

**Geometric deep learning.** Of key interest to the design of recommender systems are deep learning approaches. In the recent years, deep neural networks and, in particular, convolutional neural networks (CNNs) [25] have been applied with great success to numerous applications. However, classical CNN models cannot be directly applied to the recommendation problem to extract meaningful patterns in users, items and ratings because these data are not Euclidean structured, i.e. they do not lie on regular lattices like images but rather irregular domains like graphs. Recent works applying deep learning to recommender systems used networks with fully connected or auto-encoder architectures [44, 35, 14]. Such methods are unable to extract the important local stationary patterns from the data, which is one of the key properties of CNN architectures. New neural networks are necessary and this has motivated the recent development of *geometric deep learning* techniques that can mathematically deal with graph-structured data, which arises in numerous applications, ranging from computer graphics and vision [28, 2, 4, 3, 30] to chemistry [12]. We recommend the review paper [6] to the reader not familiar with this line of works.

The earliest attempts to apply neural networks to graphs are due to Scarselli *et al.* [13, 34] (see more recent formulation [26, 40]). Bruna *et al.* [7, 15] formulated CNN-like deep neural architectures on graphs in the spectral domain, employing the analogy between the classical Fourier transforms and projections onto the eigenbasis of the graph Laplacian operator [37]. Defferrard *et al.* [10] proposed an efficient filtering scheme using recurrent Chebyshev polynomials, which reduces the complexity of CNNs on graphs to the same complexity of classical (Euclidean) CNNs. This model was later extended to deal with dynamic data [36]. Kipf and Welling [21] proposed a simplification of Chebychev networks using simple filters operating on 1-hop neighborhoods of the graph. Monti *et al.* [30] introduced a spatial-domain generalization of CNNs to graphs local patch operators represented as Gaussian mixture models, showing significantly better generalization across different graphs.

**Contributions.** We present two main contributions. First, we introduce a new multi-graph CNN architecture that generalizes [10] to multiple graphs. This new architecture is able to extract local stationary patterns from signals defined on multiple graphs simultaneously. While in this work we apply multi-graph CNNs in the context of recommender systems to the graphs of users and items, however, our architecture is generic and can be used in other applications, such as neuroscience (autism detection with network of people and brain connectivity [31, 23]), computer graphics (shape correspondence on product manifold [41]), or social network analysis (abnormal spending behavior detection with graphs of customers and stores [39]). Second, we approach the matrix completion problem as learning on user and item graphs using the new deep multi-graph CNN framework. Our architecture is based on a cascade of multi-graph CNN followed by Long Short-Term Memory (LSTM) recurrent neural network [16] that together can be regarded as a learnable diffusion process that reconstructs the score matrix.

## 2 Background

### 2.1 Matrix Completion

**Matrix completion problem.** Recovering the missing values of a matrix given a small fraction of its entries is an ill-posed problem without additional mathematical constraints on the space of solutions. It is common to assume that the variables lie in a smaller subspace, i.e., the matrix is of low rank,

$$\min_{\mathbf{X}} \ \text{rank}(\mathbf{X}) \quad \text{s.t.} \quad x_{ij} = y_{ij}, \ \forall ij \in \Omega, \tag{1}$$

where $\mathbf{X}$ denotes the matrix to recover, $\Omega$ is the set of the known entries and $y_{ij}$ are their values. Unfortunately, rank minimization turns out to be an NP-hard combinatorial problem that is computationally intractable in practical cases. The tightest possible convex relaxation of problem (1) is to replace the rank with the nuclear norm $\| \cdot \|_{\star}$ equal to the sum of its singular values [8],

$$\min_{\mathbf{X}} \ \|\mathbf{X}\|_{\star} + \frac{\mu}{2}\|\mathbf{\Omega} \circ (\mathbf{X} - \mathbf{Y})\|_{\mathrm{F}}^2; \tag{2}$$

the equality constraint is also replaced with a penalty to make the problem more robust to noise (here $\mathbf{\Omega}$ is the indicator matrix of the known entries $\Omega$ and $\circ$ denotes the Hadamard pointwise product).

Candès and Recht [8] proved that under some technical conditions the solutions of problems (2) and (1) coincide.

**Geometric matrix completion** An alternative relaxation of the rank operator in (1) can be achieved constraining the space of solutions to be smooth w.r.t. some geometric structure on the rows and columns of the matrix [27, 19, 33, 1]. The simplest model is proximity structure represented as an undirected weighted column graph $\mathcal{G}_c = (\{1, \ldots, n\}, \mathcal{E}_c, \mathbf{W}_c)$ with *adjacency matrix* $\mathbf{W}_c = (w_{ij}^c)$, where $w_{ij}^c = w_{ji}^c$, $w_{ij}^c = 0$ if $(i,j) \notin \mathcal{E}_c$ and $w_{ij}^c > 0$ if $(i,j) \in \mathcal{E}_c$. In our setting, the column graph could be thought of as a social network capturing relations between users and the similarity of their tastes. The row graph $\mathcal{G}_r = (\{1, \ldots, m\}, \mathcal{E}_r, \mathbf{W}_r)$ representing the items similarities is defined similarly.

On each of these graphs one can construct the (normalized) *graph Laplacian*, an $n \times n$ symmetric positive-semidefinite matrix $\mathbf{\Delta} = \mathbf{I} - \mathbf{D}^{-1/2} \mathbf{W} \mathbf{D}^{-1/2}$, where $\mathbf{D} = \mathrm{diag}(\sum_{j \neq i} w_{ij})$ is the *degree matrix*. We denote the Laplacian associated with row and column graphs by $\mathbf{\Delta}_r$ and $\mathbf{\Delta}_c$, respectively. Considering the columns (respectively, rows) of matrix $\mathbf{X}$ as vector-valued functions on the column graph $\mathcal{G}_c$ (respectively, row graph $\mathcal{G}_r$), their smoothness can be expressed as the *Dirichlet norm* $\|\mathbf{X}\|_{\mathcal{G}_r}^2 = \mathrm{trace}(\mathbf{X}^\top \mathbf{\Delta}_r \mathbf{X})$ (respecitvely, $\|\mathbf{X}\|_{\mathcal{G}_c}^2 = \mathrm{trace}(\mathbf{X} \mathbf{\Delta}_c \mathbf{X}^\top)$). The *geometric matrix completion* problem [19] thus boils down to minimizing

$$\min_{\mathbf{X}} \ \|\mathbf{X}\|_{\mathcal{G}_r}^2 + \|\mathbf{X}\|_{\mathcal{G}_c}^2 + \frac{\mu}{2} \|\mathbf{\Omega} \circ (\mathbf{X} - \mathbf{Y})\|_{\mathrm{F}}^2. \tag{3}$$

**Factorized models.** Matrix completion algorithms introduced in the previous section are well-posed as convex optimization problems, guaranteeing existence, uniqueness and robustness of solutions. Besides, fast algorithms have been developed for the minimization of the non-differentiable nuclear norm. However, the variables in this formulation are the full $m \times n$ matrix $\mathbf{X}$, making it hard to scale up to large matrices such as the Netflix challenge.

A solution is to use a factorized representation [38, 22, 27, 43, 33, 1] $\mathbf{X} = \mathbf{W}\mathbf{H}^\top$, where $\mathbf{W}, \mathbf{H}$ are $m \times r$ and $n \times r$ matrices, respectively, with $r \ll \min(m, n)$. The use of factors $\mathbf{W}, \mathbf{H}$ reduces the number of degrees of freedom from $\mathcal{O}(mn)$ to $\mathcal{O}(m+n)$; this representation is also attractive as people often assumes the original matrix to be low-rank for solving the matrix completion problem, and $\mathrm{rank}(\mathbf{W}\mathbf{H}^\top) \leq r$ by construction.

The nuclear norm minimization problem (2) can be rewritten in a factorized form as [38]:

$$\min_{\mathbf{W}, \mathbf{H}} \ \frac{1}{2}\|\mathbf{W}\|_{\mathrm{F}}^2 + \frac{1}{2}\|\mathbf{H}\|_{\mathrm{F}}^2 + \frac{\mu}{2} \|\mathbf{\Omega} \circ (\mathbf{W}\mathbf{H}^\top - \mathbf{Y})\|_{\mathrm{F}}^2. \tag{4}$$

and the factorized formulation of the graph-based minimization problem (3) as

$$\min_{\mathbf{W}, \mathbf{H}} \ \frac{1}{2}\|\mathbf{W}\|_{\mathcal{G}_r}^2 + \frac{1}{2}\|\mathbf{H}\|_{\mathcal{G}_c}^2 + \frac{\mu}{2} \|\mathbf{\Omega} \circ (\mathbf{W}\mathbf{H}^\top - \mathbf{Y})\|_{\mathrm{F}}^2. \tag{5}$$

The limitation of model (5) is that it decouples the regularization previously applied simultaneously on the rows and columns of $\mathbf{X}$ in (3), but the advantage is linear instead of quadratic complexity.

## 2.2 Deep learning on graphs

The key concept underlying our work is *geometric deep learning*, an extension of CNNs to graphs. In particular, we focus here on graph CNNs formulated in the spectral domain. A graph Laplacian admits a spectral eigendecomposition of the form $\mathbf{\Delta} = \mathbf{\Phi} \mathbf{\Lambda} \mathbf{\Phi}^\top$, where $\mathbf{\Phi} = (\boldsymbol{\phi}_1, \ldots \boldsymbol{\phi}_n)$ denotes the matrix of orthonormal eigenvectors and $\mathbf{\Lambda} = \mathrm{diag}(\lambda_1, \ldots, \lambda_n)$ is the diagonal matrix of the corresponding eigenvalues. The eigenvectors play the role of Fourier atoms in classical harmonic analysis and the eigenvalues can be interpreted as frequencies. Given a function $\mathbf{x} = (x_1, \ldots, x_n)^\top$ on the vertices of the graph, its *graph Fourier transform* is given by $\hat{\mathbf{x}} = \mathbf{\Phi}^\top \mathbf{x}$. The *spectral convolution* of two functions $\mathbf{x}, \mathbf{y}$ can be defined as the element-wise product of the respective Fourier transforms,

$$\mathbf{x} \star \mathbf{y} = \mathbf{\Phi}(\mathbf{\Phi}^\top \mathbf{y}) \circ (\mathbf{\Phi}^\top \mathbf{x}) = \mathbf{\Phi} \, \mathrm{diag}(\hat{y}_1, \ldots, \hat{y}_n) \hat{\mathbf{x}}, \tag{6}$$

by analogy to the Convolution Theorem in the Euclidean case.

Bruna *et al*. [7] used the spectral definition of convolution (6) to generalize CNNs on graphs. A spectral convolutional layer in this formulation has the form

$$\tilde{\mathbf{x}}_l = \xi \left( \sum_{l'=1}^{q'} \mathbf{\Phi} \hat{\mathbf{Y}}_{ll'} \mathbf{\Phi}^\top \mathbf{x}_{l'} \right), \quad l = 1, \ldots, q, \tag{7}$$

where $q', q$ denote the number of input and output channels, respectively, $\hat{\mathbf{Y}}_{ll'} = \mathrm{diag}(\hat{y}_{ll',1}, \ldots, \hat{y}_{ll',n})$ is a diagonal matrix of spectral multipliers representing a learnable filter in the spectral domain, and $\xi$ is a nonlinearity (e.g. ReLU) applied on the vertex-wise function values. Unlike classical convolutions carried out efficiently in the spectral domain using FFT, the computations of the forward and inverse graph Fourier transform incur expensive $\mathcal{O}(n^2)$ multiplication by the matrices $\mathbf{\Phi}, \mathbf{\Phi}^\top$, as there are no FFT-like algorithms on general graphs. Second, the number of parameters representing the filters of each layer of a spectral CNN is $\mathcal{O}(n)$, as opposed to $\mathcal{O}(1)$ in classical CNNs. Third, there is no guarantee that the filters represented in the spectral domain are localized in the spatial domain, which is another important property of classical CNNs.

Henaff *et al.* [15] argued that spatial localization can be achieved by forcing the spectral multipliers to be smooth. The filter coefficients are represented as $\hat{y}_k = \tau(\lambda_k)$, where $\tau(\lambda)$ is a smooth transfer function of frequency $\lambda$; its application to a signal $\mathbf{x}$ is expressed as $\tau(\mathbf{\Delta})\mathbf{x} = \mathbf{\Phi}\,\mathrm{diag}(\tau(\lambda_1), \ldots, \tau(\lambda_n))\mathbf{\Phi}^\top \mathbf{x}$, where applying a function to a matrix is understood in the operator sense and boils down to applying the function to the matrix eigenvalues. In particular, the authors used parametric filters of the form

$$\tau_{\boldsymbol{\theta}}(\lambda) = \sum_{j=1}^{p} \theta_j \beta_j(\lambda), \tag{8}$$

where $\beta_1(\lambda), \ldots, \beta_r(\lambda)$ are some fixed interpolation kernels, and $\boldsymbol{\theta} = (\theta_1, \ldots, \theta_p)$ are $p = \mathcal{O}(1)$ interpolation coefficients acting as parameters of the spectral convolutional layer.

Defferrard *et al.* [10] used polynomial filters of order $p$ represented in the Chebyshev basis,

$$\tau_{\boldsymbol{\theta}}(\tilde{\lambda}) = \sum_{j=0}^{p} \theta_j T_j(\tilde{\lambda}), \tag{9}$$

where $\tilde{\lambda}$ is frequency rescaled in $[-1, 1]$, $\boldsymbol{\theta}$ is the $(p{+}1)$-dimensional vector of polynomial coefficients parametrizing the filter, and $T_j(\lambda) = 2\lambda T_{j-1}(\lambda) - T_{j-2}(\lambda)$ denotes the Chebyshev polynomial of degree $j$ defined in a recursive manner with $T_1(\lambda) = \lambda$ and $T_0(\lambda) = 1$. Here, $\tilde{\mathbf{\Delta}} = 2\lambda_n^{-1}\mathbf{\Delta} - \mathbf{I}$ is the rescaled Laplacian with eigenvalues $\tilde{\mathbf{\Lambda}} = 2\lambda_n^{-1}\mathbf{\Lambda} - \mathbf{I}$ in the interval $[-1, 1]$.

This approach benefits from several advantages. First, it does not require an explicit computation of the Laplacian eigenvectors, as applying a Chebyshev filter to $\mathbf{x}$ amounts to $\tau_{\boldsymbol{\theta}}(\tilde{\mathbf{\Delta}})\mathbf{x} = \sum_{j=0}^{p} \theta_j T_j(\tilde{\mathbf{\Delta}})\mathbf{x}$; due to the recursive definition of the Chebyshev polynomials, this incurs applying the Laplacian $p$ times. Multiplication by Laplacian has the cost of $\mathcal{O}(|\mathcal{E}|)$, and assuming the graph has $|\mathcal{E}| = \mathcal{O}(n)$ edges (which is the case for $k$-nearest neighbors graphs and most real-world networks), the overall complexity is $\mathcal{O}(n)$ rather than $\mathcal{O}(n^2)$ operations, similarly to classical CNNs. Moreover, since the Laplacian is a local operator affecting only 1-hop neighbors of a vertex and accordingly its $p$th power affects the $p$-hop neighborhood, the resulting filters are spatially localized.

## 3 Our approach

In this paper, we propose formulating matrix completion as a problem of deep learning on user and item graphs. We consider two architectures summarized in Figures 1 and 2. The first architecture works on the full matrix model producing better accuracy but requiring higher complexity. The second architecture used factorized matrix model, offering better scalability at the expense of slight reduction of accuracy. For both architectures, we consider a combination of multi-graph CNN and RNN, which will be described in detail in the following sections. Multi-graph CNNs are used to extract local stationary features from the score matrix using row and column similarities encoded by user and item graphs. Then, these spatial features are fed into a RNN that diffuses the score values progressively, reconstructing the matrix.

### 3.1 Multi-Graph CNNs

**Multi-graph convolution.** Our first goal is to extend the notion of the aforementioned graph Fourier transform to matrices whose rows and columns are defined on row- and column-graphs. We recall that the classical two-dimensional Fourier transform of an image (matrix) can be thought of as applying a one-dimensional Fourier transform to its rows and columns. In our setting, the analogy of the two-dimensional Fourier transform has the form

$$\hat{\mathbf{X}} = \mathbf{\Phi}_r^\top \mathbf{X} \mathbf{\Phi}_c \tag{10}$$

where $\mathbf{\Phi}_c, \mathbf{\Phi}_r$ and $\mathbf{\Lambda}_c = \mathrm{diag}(\lambda_{c,1}, \ldots, \lambda_{c,n})$ and $\mathbf{\Lambda}_r = \mathrm{diag}(\lambda_{r,1}, \ldots, \lambda_{r,m})$ denote the $n \times n$ and $m \times m$ eigenvector- and eigenvalue matrices of the column- and row-graph Laplacians $\mathbf{\Delta}_c, \mathbf{\Delta}_r$, respectively. The multi-graph version of the spectral convolution (6) is given by

$$\mathbf{X} \star \mathbf{Y} = \mathbf{\Phi}_r(\hat{\mathbf{X}} \circ \hat{\mathbf{Y}})\mathbf{\Phi}_c^\top, \tag{11}$$

and in the classical setting can be thought as the analogy of filtering a 2D image in the spectral domain (column and row graph eigenvalues $\lambda_c$ and $\lambda_r$ generalize the $x$- and $y$-frequencies of an image).

As in [7], representing multi-graph filters as their spectral multipliers $\hat{\mathbf{Y}}$ would yield $\mathcal{O}(mn)$ parameters, prohibitive in any practical application. To overcome this limitation, we follow [15], assuming that the multi-graph filters are expressed in the spectral domain as a smooth function of both frequencies (eigenvalues $\lambda_c, \lambda_r$ of the row- and column graph Laplacians) of the form $\hat{\mathbf{Y}}_{k,k'} = \tau(\lambda_{c,k}, \lambda_{r,k'})$. In particular, using Chebychev polynomial filters of degree $p$,[1]

$$\tau_{\mathbf{\Theta}}(\tilde{\lambda}_c, \tilde{\lambda}_r) = \sum_{j,j'=0}^{p} \theta_{jj'} T_j(\tilde{\lambda}_c) T_{j'}(\tilde{\lambda}_r), \tag{12}$$

where $\tilde{\lambda}_c, \tilde{\lambda}_r$ are the frequencies rescaled $[-1, 1]$ (see Figure 4 for examples). Such filters are parametrized by a $(p+1) \times (p+1)$ matrix of coefficients $\mathbf{\Theta} = (\theta_{jj'})$, which is $\mathcal{O}(1)$ in the input size as in classical CNNs on images. The application of a multi-graph filter to the matrix $\mathbf{X}$

$$\tilde{\mathbf{X}} = \sum_{j,j'=0}^{p} \theta_{jj'} T_j(\tilde{\mathbf{\Delta}}_r) \mathbf{X} T_{j'}(\tilde{\mathbf{\Delta}}_c) \tag{13}$$

incurs an $\mathcal{O}(mn)$ computational complexity (here, as previously, $\tilde{\mathbf{\Delta}}_c = 2\lambda_{c,n}^{-1}\mathbf{\Delta}_c - \mathbf{I}$ and $\tilde{\mathbf{\Delta}}_r = 2\lambda_{r,m}^{-1}\mathbf{\Delta}_r - \mathbf{I}$ denote the scaled Laplacians).

Similarly to (7), a multi-graph convolutional layer using the parametrization of filters according to (13) is applied to $q'$ input channels ($m \times n$ matrices $\mathbf{X}_1, \ldots, \mathbf{X}_{q'}$ or a tensor of size $m \times n \times q'$),

$$\tilde{\mathbf{X}}_l = \xi\left(\sum_{l'=1}^{q'} \mathbf{X}_{l'} \star \mathbf{Y}_{ll'}\right) = \xi\left(\sum_{l'=1}^{q'} \sum_{j,j'=0}^{p} \theta_{jj',ll'} T_j(\tilde{\mathbf{\Delta}}_r) \mathbf{X}_{l'} T_{j'}(\tilde{\mathbf{\Delta}}_c)\right), \quad l = 1, \ldots, q, \tag{14}$$

producing $q$ outputs (tensor of size $m \times n \times q$). Several layers can be stacked together. We call such an architecture a *Multi-Graph CNN* (MGCNN).

**Separable convolution.** A simplification of the multi-graph convolution is obtained considering the factorized form of the matrix $\mathbf{X} = \mathbf{W}\mathbf{H}^\top$ and applying one-dimensional convolutions to the respective graph to each factor. Similarly to the previous case, we can express the filters resorting to Chebyshev polynomials,

$$\tilde{\mathbf{w}}_l = \sum_{j=0}^{p} \theta_j^r T_j(\tilde{\mathbf{\Delta}}_r) \mathbf{w}_l, \qquad \tilde{\mathbf{h}}_l = \sum_{j'=0}^{p} \theta_{j'}^c T_{j'}(\tilde{\mathbf{\Delta}}_c) \mathbf{h}_l, \qquad l = 1, \ldots, r \tag{15}$$

where $\mathbf{w}_l, \mathbf{h}_l$ denote the $l$th columns of factors $\mathbf{W}, \mathbf{H}$ and $\boldsymbol{\theta}^r = (\theta_0^r, \ldots, \theta_p^r)$ and $\boldsymbol{\theta}^c = (\theta_0^c, \ldots, \theta_p^c)$ are the parameters of the row- and column- filters, respectively (a total of $2(p+1) = \mathcal{O}(1)$). Application of such filters to $\mathbf{W}$ and $\mathbf{H}$ incurs $\mathcal{O}(m+n)$ complexity. Convolutional layers (14) thus take the form

$$\tilde{\mathbf{w}}_l = \xi\left(\sum_{l'=1}^{q'} \sum_{j=0}^{p} \theta_{j,ll'}^r T_j(\tilde{\mathbf{\Delta}}_r) \mathbf{w}_{l'}\right), \qquad \tilde{\mathbf{h}}_l = \xi\left(\sum_{l'=1}^{q'} \sum_{j'=0}^{p} \theta_{j',ll'}^c T_{j'}(\tilde{\mathbf{\Delta}}_c) \mathbf{h}_{l'}\right). \tag{16}$$

We call such an architecture a *separable MGCNN* or *sMGCNN*.

## 3.2 Matrix diffusion with RNNs

The next step of our approach is to feed the spatial features extracted from the matrix by the MGCNN or sMGCNN to a recurrent neural network (RNN) implementing a diffusion process that progressively reconstructs the score matrix (see Figure 3). Modelling matrix completion as a diffusion process

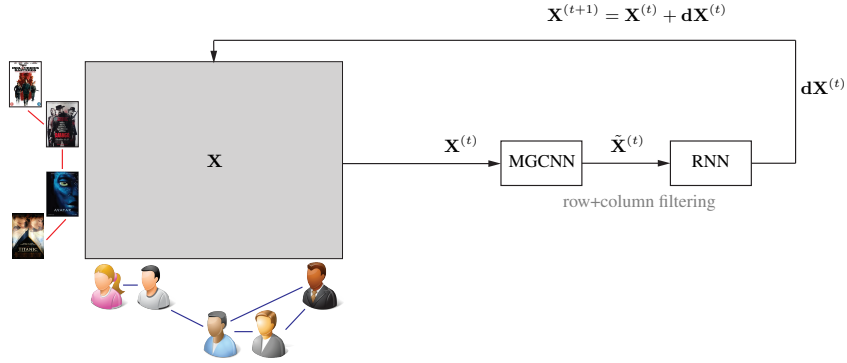

Figure 1: Recurrent MGCNN (RMGCNN) architecture using the full matrix completion model and operating simultaneously on the rows and columns of the matrix $\mathbf{X}$. Learning complexity is $\mathcal{O}(mn)$.

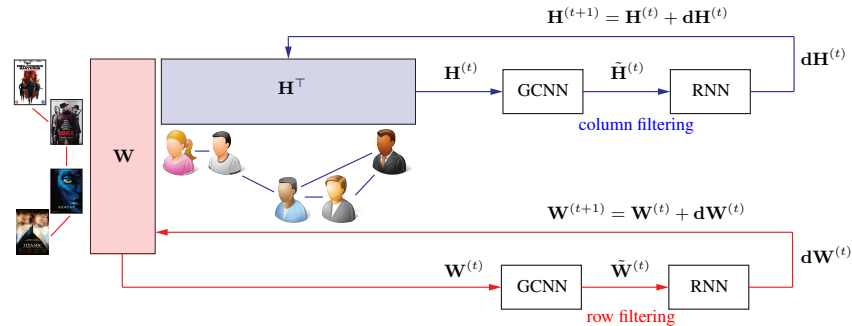

Figure 2: Separable Recurrent MGCNN (sRMGCNN) architecture using the factorized matrix completion model and operating separately on the rows and columns of the factors $\mathbf{W}, \mathbf{H}^{\top}$. Learning complexity is $\mathcal{O}(m + n)$.

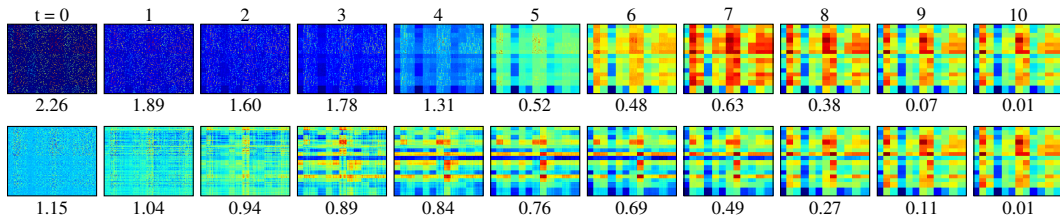

Figure 3: Evolution of matrix $\mathbf{X}^{(t)}$ with our architecture using full matrix completion model RMGCNN (top) and factorized matrix completion model sRMGCNN (bottom). Numbers indicate the RMS error.

appears particularly suitable for realizing an architecture which is independent of the sparsity of the available information. In order to combine the few scores available in a sparse input matrix, a multilayer CNN would require very large filters or many layers to diffuse the score information across matrix domains. On the contrary, our diffusion-based approach allows to reconstruct the missing information just by imposing the proper amount of diffusion iterations. This gives the possibility to deal with extremely sparse data, without requiring at the same time excessive amounts of model parameters. See Table 3 for an experimental evaluation on this aspect.

We use the classical LSTM architecture [16], which has demonstrated to be highly efficient to learn complex non-linear diffusion processes due to its ability to keep long-term internal states (in particular, limiting the vanishing gradient issue). The input of the LSTM gate is given by the static features extracted from the MGCNN, which can be seen as a projection or dimensionality reduction of the original matrix in the space of the most meaningful and representative information (the disentanglement effect). This representation coupled with LSTM appears particularly well-suited to keep a long term internal state, which allows to predict accurate small changes $\mathbf{dX}$ of the matrix $\mathbf{X}$ (or $\mathbf{dW}, \mathbf{dH}$ of the factors $\mathbf{W}, \mathbf{H}$) that can propagate through the full temporal steps.

Figures 1 and 2 and Algorithms 1 and 2 summarize the proposed matrix completion architectures. We refer to the whole architecture combining the MGCNN and RNN in the full matrix completion setting as *recurrent multi-graph CNN (RMGCNN)*. The factorized version with separable MGCNN and RNN is referred to as *separable RMGCNN (sRMGCNN)*. The complexity of Algorithm 1 scales quadratically as $\mathcal{O}(mn)$ due to the use of MGCNN. For large matrices, Algorithm 2 that processes the rows and columns separately with standard GCNNs and scales linearly as $\mathcal{O}(m + n)$ is preferable.

We will demonstrate in Section 4 that the proposed RMGCNN and sRMGCNN architectures show themselves very well on different settings of matrix completion problems. However, we should note that this is just one possible configuration, which we by no means claim to be optimal. For example, in all our experiments we used only one convolutional layer; it is likely that better yet performance could be achieved with more layers.

---

**Algorithm 1 (RMGCNN)**

**input** $m \times n$ matrix $\mathbf{X}^{(0)}$ containing initial values

1: **for** $t = 0 : T$ **do**
2:     Apply the Multi-Graph CNN (13) on $\mathbf{X}^{(t)}$ producing an $m \times n \times q$ output $\tilde{\mathbf{X}}^{(t)}$.
3:     **for** all elements $(i, j)$ **do**
4:         Apply RNN to $q$-dim $\tilde{\mathbf{x}}_{ij}^{(t)} = (\tilde{x}_{ij1}^{(t)}, \ldots, \tilde{x}_{ijq}^{(t)})$ producing incremental update $dx_{ij}^{(t)}$
5:     **end for**
6:     Update $\mathbf{X}^{(t+1)} = \mathbf{X}^{(t)} + \mathbf{dX}^{(t)}$
7: **end for**

---

**Algorithm 2 (sRMGCNN)**

**input** $m \times r$ factor $\mathbf{H}^{(0)}$ and $n \times r$ factor $\mathbf{W}^{(0)}$ representing the matrix $\mathbf{X}^{(0)}$

1: **for** $t = 0 : T$ **do**
2:     Apply the Graph CNN on $\mathbf{H}^{(t)}$ producing an $n \times q$ output $\tilde{\mathbf{H}}^{(t)}$.
3:     **for** $j = 1 : n$ **do**
4:         Apply RNN to $q$-dim $\tilde{\mathbf{h}}_{j}^{(t)} = (\tilde{h}_{j1}^{(t)}, \ldots, \tilde{h}_{jq}^{(t)})$ producing incremental update $dh_{j}^{(t)}$
5:     **end for**
6:     Update $\mathbf{H}^{(t+1)} = \mathbf{H}^{(t)} + \mathbf{dH}^{(t)}$
7:     Repeat steps 2-6 for $\mathbf{W}^{(t+1)}$
8: **end for**

---

### 3.3 Training

Training of the networks is performed by minimizing the loss

$$\ell(\boldsymbol{\Theta}, \boldsymbol{\sigma}) = \|\mathbf{X}_{\boldsymbol{\Theta}, \boldsymbol{\sigma}}^{(T)}\|_{\mathcal{G}_r}^2 + \|\mathbf{X}_{\boldsymbol{\Theta}, \boldsymbol{\sigma}}^{(T)}\|_{\mathcal{G}_c}^2 + \frac{\mu}{2}\|\boldsymbol{\Omega} \circ (\mathbf{X}_{\boldsymbol{\Theta}, \boldsymbol{\sigma}}^{(T)} - \mathbf{Y})\|_{\mathrm{F}}^2. \tag{17}$$

Here, $T$ denotes the number of diffusion iterations (applications of the RNN), and we use the notation $\mathbf{X}_{\boldsymbol{\Theta}, \boldsymbol{\sigma}}^{(T)}$ to emphasize that the matrix depends on the parameters of the MGCNN (Chebyshev polynomial coefficients $\boldsymbol{\Theta}$) and those of the LSTM (denoted by $\boldsymbol{\sigma}$). In the factorized setting, we use the loss

$$\ell(\boldsymbol{\theta}_r, \boldsymbol{\theta}_c, \boldsymbol{\sigma}) = \|\mathbf{W}_{\boldsymbol{\theta}_r, \boldsymbol{\sigma}}^{(T)}\|_{\mathcal{G}_r}^2 + \|\mathbf{H}_{\boldsymbol{\theta}_c, \boldsymbol{\sigma}}^{(T)}\|_{\mathcal{G}_c}^2 + \frac{\mu}{2}\|\boldsymbol{\Omega} \circ (\mathbf{W}_{\boldsymbol{\theta}_r, \boldsymbol{\sigma}}^{(T)}(\mathbf{H}_{\boldsymbol{\theta}_c, \boldsymbol{\sigma}}^{(T)})^{\top} - \mathbf{Y})\|_{\mathrm{F}}^2 \tag{18}$$

where $\boldsymbol{\theta}_c, \boldsymbol{\theta}_r$ are the parameters of the two GCNNs.

## 4 Results[2]

**Experimental settings.** We closely followed the experimental setup of [33], using five standard datasets: Synthetic dataset from [19], MovieLens [29], Flixster [18], Douban [27], and YahooMusic [11]. We used disjoint training and test sets and the presented results are reported on test sets in all our experiments. As in [33], we evaluated MovieLens using only the first of the 5 provided data splits. For Flixster, Douban and YahooMusic, we evaluated on a reduced matrix of 3000 users and items, considering 90% of the given scores as training set and the remaining as test set. Classical Matrix Completion (MC) [9], Inductive Matrix Completion (IMC) [17, 42], Geometric Matrix Completion (GMC) [19], and Graph Regularized Alternating Least Squares (GRALS) [33] were used as baseline methods. In all the experiments, we used the following settings for our RMGCNNs: Chebyshev polynomials of order $p = 4$, outputting $k = 32$-dimensional features, LSTM cells with 32 features and $T = 10$ diffusion steps (for both training and test). The number of diffusion steps $T$ has been estimated on the Movielens validation set and used in all our experiments. A better estimate of $T$ can be done by cross-validation, and thus can potentially only improve the final results. All the

models were implemented in Google TensorFlow and trained using the Adam stochastic optimization algorithm [20] with learning rate $10^{-3}$. In factorized models, ranks $r = 15$ and 10 was used for the synthetic and real datasets, respectively. For all methods, hyperparameters were chosen by cross-validation.

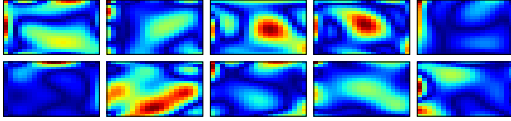

Figure 4: Absolute value $|\tau(\tilde{\lambda}_c, \tilde{\lambda}_r)|$ of the first ten spectral filters learnt by our MGCNN model. In each matrix, rows and columns represent frequencies $\tilde{\lambda}_r$ and $\tilde{\lambda}_c$ of the row and column graphs, respectively.

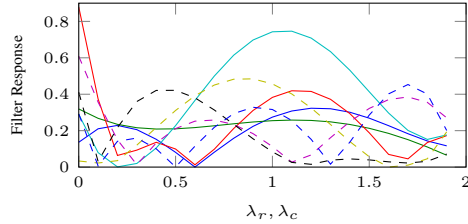

Figure 5: Absolute values $|\tau(\tilde{\lambda}_c)|$ and $|\tau(\tilde{\lambda}_r)|$ of the first four column (solid) and row (dashed) spectral filters learned by our sMGCNN model.

## 4.1 Synthetic data

We start the experimental evaluation showing the performance of our approach on a small synthetic dataset, in which the user and item graphs have strong communities structure. Though rather simple, such a dataset allows to study the behavior of different algorithms in controlled settings.

The performance of different matrix completion methods is reported in Table 1, along with their theoretical complexity. Our RMGCNN and sRMGCNN models achieve better accuracy than other methods with lower complexity. Different diffusion time steps of these two models are visualized in Figure 3. Figures 4 and 5 depict the spectral filters learnt by MGCNN and row- and column-GCNNs.

We repeated the same experiment assuming only the column (users) graph to be given. In this setting, RMGCNN cannot be applied, while sRMGCNN has only one GCNN applied on the factor **H** (the other factor **W** is free). Table 2 summarizes the results of this experiment, again, showing that our approach performs the best.

Table 3 compares our RMGCNN with more classical multilayer MGCNNs. Our recurrent solutions outperforms deeper and more complex architectures, requiring at the same time a lower amount of parameters.

Table 1: Comparison of different matrix completion methods using *users+items graphs* in terms of number of parameters (optimization variables) and computational complexity order (operations per iteration). Big-O notation is avoided for clarity reasons. Rightmost column shows the RMS error on Synthetic dataset.

| METHOD | PARAMS | NO. OP. | RMSE |
|---|---|---|---|
| GMC | $mn$ | $mn$ | 0.3693 |
| GRALS | $m+n$ | $m+n$ | 0.0114 |
| **sRMGCNN** | **1** | $\boldsymbol{m+n}$ | **0.0106** |
| **RMGCNN** | **1** | $\boldsymbol{mn}$ | **0.0053** |

Table 2: Comparison of different matrix completion methods using *users graph only* in terms of number of parameters (optimization variables) and computational complexity order (operations per iteration). Big-O notation is avoided for clarity reasons. Rightmost column shows the RMS error on Synthetic dataset.

| METHOD | PARAMS | NO. OP. | RMSE |
|---|---|---|---|
| GRALS | $m+n$ | $m+n$ | 0.0452 |
| **sRMGCNN** | $\boldsymbol{m}$ | $\boldsymbol{m+n}$ | **0.0362** |

Table 3: Reconstruction errors for the synthetic dataset between multiple convolutional layers architectures and the proposed architecture. Chebyshev polynomials of order 4 have been used for both users and movies graphs ($q'$MGC$q$ denotes a multi-graph convolutional layer with $q'$ input features and $q$ output features).

| Method | Params | Architecture | RMSE |
|---|---|---|---|
| MGCNN$_{3\text{layers}}$ | $9K$ | 1MGC32, 32MGC10, 10MGC1 | 0.0116 |
| MGCNN$_{4\text{layers}}$ | $53K$ | 1MGC32, 32MGC32 $\times$ 2, 32MGC1 | 0.0073 |
| MGCNN$_{5\text{layers}}$ | $78K$ | 1MGC32, 32MGC32 $\times$ 3, 32MGC1 | 0.0074 |
| MGCNN$_{6\text{layers}}$ | $104K$ | 1MGC32, 32MGC32 $\times$ 4, 32MGC1 | 0.0064 |
| **RMGCNN** | **9K** | 1MGC32 + LSTM | **0.0053** |

## 4.2 Real data

Following [33], we evaluated the proposed approach on the MovieLens, Flixster, Douban and YahooMusic datasets. For the MovieLens dataset we constructed the user and item (movie) graphs as unweighted 10-nearest neighbor graphs in the space of user and movie features, respectively. For Flixster, the user and item graphs were constructed from the scores of the original matrix. On this dataset, we also performed an experiment using only the users graph. For the Douban dataset, we used only the user graph (provided in the form of a social network). For the YahooMusic dataset, we used only the item graph, constructed with unweighted 10-nearest neighbors in the space of item features (artists, albums, and genres). For the latter three datasets, we used a sub-matrix of $3000 \times 3000$ entries for evaluating the performance. Tables 4 and 5 summarize the performance of different methods. sRMGCNN outperforms the competitors in all the experiments.

Table 4: Performance (RMS error) of different matrix completion methods on the MovieLens dataset.

| METHOD | RMSE |
|---|---|
| GLOBAL MEAN | 1.154 |
| USER MEAN | 1.063 |
| MOVIE MEAN | 1.033 |
| MC [9] | 0.973 |
| IMC [17, 42] | 1.653 |
| GMC [19] | 0.996 |
| GRALS [33] | 0.945 |
| **sRMGCNN** | **0.929** |

Table 5: Performance (RMS error) on several datasets. For Douban and YahooMusic, a single graph (of users and items respectively) was used. For Flixster, two settings are shown: users+items graphs / only users graph.

| METHOD | FLIXSTER | DOUBAN | YAHOOMUSIC |
|---|---|---|---|
| GRALS | 1.3126 / 1.2447 | 0.8326 | 38.0423 |
| **sRMGCNN** | **1.1788 / 0.9258** | **0.8012** | **22.4149** |

## 5 Conclusions

In this paper, we presented a new deep learning approach for matrix completion based on multi-graph convolutional neural network architecture. Among the key advantages of our approach compared to traditional methods is its low computational complexity and constant number of degrees of freedom independent of the matrix size. We showed that the use of deep learning for matrix completion allows to beat related state-of-the-art recommender system methods. To our knowledge, our work is the first application of deep learning on graphs to this class of problems. We believe that it shows the potential of the nascent field of geometric deep learning on non-Euclidean domains, and will encourage future works in this direction.

**Acknowledgments**

FM and MB are supported in part by ERC Starting Grant No. 307047 (COMET), ERC Consolidator Grant No. 724228 (LEMAN), Google Faculty Research Award, Nvidia equipment grant, Radcliffe fellowship from Harvard Institute for Advanced Study, and TU Munich Institute for Advanced Study, funded by the German Excellence Initiative and the European Union Seventh Framework Programme under grant agreement No. 291763. XB is supported in part by NRF Fellowship NRFF2017-10.

## Footnotes

[1]For simplicity, we use the same degree $p$ for row- and column frequencies.

[2]Code: https://github.com/fmonti/mgcnn

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
