[Reviews · NeurIPS 2017]

Reviewer 1



This paper generalizes a recent NIPS 2016 paper [10] by allowing convolutional neural network (CNN) models to work on multiple graphs. It extracts local stationary patterns from signals defined on the graphs simultaneously. In particular, it is applied to recommender systems by considering the graphs defined between users and between items. The multi-graph CNN model is followed by a recurrent neural network (RNN) with long short-term memory (LSTM) cells to complete the score matrix. Strengths of the paper: * The proposed deep learning architecture is novel for solving the matrix completion problem in recommender systems with the relationships between users and the relationships between items represented as two graphs. * The proposed method has relatively low computational complexity. * The proposed method has good performance in terms of accuracy. Weaknesses of the paper: * The multi-graph CNN part of the proposed method is explained well, but the RNN part leaves open many questions. Apart from illustrating the process using an example in Figure 3, there is not much discussion to explain why it can be regarded as a matrix diffusion process. Does the process correspond to optimizing with respect to some criterion? What is the effect of changing the number of diffusion steps on the completed matrix and how is the number determined? This needs more detailed elaboration. In addition, to demonstrate that incorporating the pairwise relationships between users or items helps, you may add some degenerate versions of RMGCNN or sRMGCNN by removing one or both graphs while keeping the rest of the architecture unchanged (including the RNN part). Minor comments (just some examples of the language errors): #18: “Two major approach” #32: “makes well-defined” #66: “Long-Short Term Memory” #76: “turns out an NP-hard combinatorial problem” #83-84: “to constraint the space of solutions” #104-105: “this representation … often assumes” #159: “in details” Table 1 & Table 2: strictly speaking you should show the number of parameters and complexity using the big-O notation. The reference list contains many arXiv papers. For those papers that have already been accepted, the proper sources should be listed instead.

Reviewer 2



This paper studies matrix completion problem when side information about users and items is given that can be exploited in recommendation. Unlike existing methods such as induced matrix completion or geometric matrix completion that tries to explicitly exploit the auxiliary information in factorization methods to reduce the sample complexity to mitigate for data sparsity, this paper introduces a multi-graph convolutional neural network to complete the partially observed matrix. The authors conducted through experiments on synthetic and real datasets and compared the proposed algorithm to existing methods that demonstrates the effectiveness of proposed method. The presentation of the paper was mostly clear. The claimed contributions are discussed in the light of existing results and the paper does survey related work appropriately. Regarding the quality of the writing, the paper is reasonably well written, the structure and language are good. Overall, on the positive side, the paper has nice results, proposes an interesting algorithm, and experiments are thorough and compelling. But the paper lacks enough novelty, and in my opinion the main contribution of this paper is the generalization of recently proposed convolutional neural networks on a single graph to multiple graphs, not particularly interesting.

Reviewer 3



In short --- Paper --- The paper introduces a Neural Network (NN) architecture which targets matrix completion with graph-based constraints on rows and columns of the matrix. The proposed architecture builds upon graph convolutional NN (which are extended to multi-graph setting) and recurrent NN. The efficiency of that architecture is demonstrated on standard data-sets. --- Review --- The paper is clear and complete, and the proposed architecture is interesting and efficient on the considered datasets.